# Performance Comparison of High-Speed Motors for Electric Vehicle

**Kohei Aiso** [1,*] and **Kan Akatsu** [2]

1   Shibaura Institute of Technology, Tokyo 135-8548, Japan
2   Department of Mathematics, Physics, Electrical Engineering and Computer Science,
    Yokohama National University, Yokohama 240-8501, Japan; akatsu-kan-py@ynu.ac.jp
*   Correspondence: k-aiso@shibaura-it.ac.jp

**Abstract:** It is predicted that the maximum speed of EV traction motors will increase in the future due to reductions in size and weight. The high-speed motors are required to have high mechanical strength of the rotor for high-speed rotation, in addition to satisfying the required output and high efficiency in the wide operation area. Therefore, it is necessary to evaluate the advantages and disadvantages of motors in terms of both electrical and mechanical points of view. In this research, three motor types, PMSM, SRM, and IM, which targeted the output power of 85 kW and the maximum speed of 52,000 min$^{-1}$, are designed for use with EV traction motors, and the study clarifies which the type of motor is most suitable for application in high-speed motors of EVs in terms of their mechanical and electrical characteristics.

**Keywords:** electric vehicle; high-speed motor; permanent magnet synchronous motor; switched reluctance motor; induction motor





## 1. Introduction

Electric vehicles (EVs) have become popular as an energy-saving countermeasure. EV traction motors are required to have high efficiency, high power density, low manufacturing costs, and downsized volume. Recently, downsized motors in particular have been needed to improve electric power consumption levels and retain space where the motor is housed. One of the methods used to downsize motor volume is to increase the motor speed [1]. The output power of motors is defined by the product of the torque and the rotation speed. Downsizing can be realized by increasing the rotation speed while obtaining the same output power. For high-speed drives, motors need to achieve adequate mechanical and electrical performances (i.e., high mechanical strength of the rotor and a reduction in loss).

Permanent magnet synchronous motors (PMSMs), which use rare earth magnets, are widely used as traction motors since they can obtain high torque density and high efficiency [2,3]. However, PMSMs have several disadvantages such as high manufacturing costs and low mechanical strength due to the rare earth permanent magnets used in the rotor. These magnets are broken by centrifugal force, since high Mises stress is concentrated to these magnets in high-speed drive. Although interior permanent magnet synchronous motors (IPMSMs) prevent these magnets from breaking, high Mises stress is generated at the edge of the flux barrier via high-speed rotation. Therefore, the advanced optimization of the detailed rotor shape is required in the design process to reduce the Mises stress [4]. Meanwhile, surface permanent magnet synchronous motors (SPMSMs) can reduce the Mises stress more effectively compared with IPMSMs, and the rotor can be reinforced by using reinforcing materials such as carbon fiber and titanium alloy [5,6]. However, it requires a complex and high-cost structure. Additionally, it needs to energize the d-axis current by the flux weakening control in the high-speed region since the back electromotive force increases due to the magnet of rotor, which results in the generation of copper loss. Moreover, the efficiency of PMSMs is generally decreased in the high-speed region due

to the eddy current loss of the magnet, and both a retaining sleeve in the rotor and a segmentation magnet are required [7].

Switched reluctance motors (SRMs) have been put forward as candidate automotive motors. SRMs have a salient pole structure in the stator and the rotor and only consist of the laminated core and the winding. Therefore, SRMs have a robust structure and can reduce the Mises stress generated in the rotor (as they have no magnets). They are suitable for use in the high-speed region compared with PMSMs since SRMs not only have high mechanical strength but also no back electromotive force that is caused by magnets in a rotor. However, the torque density of SRMs is lower than that of PMSMs. Therefore, the ampere turns and the motor volume increase to achieve the same torque as PMSMs. Moreover, SRMs decrease the motor efficiency since the iron loss increases in the high-speed drive. To overcome these disadvantages, SRMs using low-iron-loss steel, 0.1 mm thick high-silicon steel, and amorphous steel sheets was proposed to achieve high efficiency in the high-speed region [8,9].

Induction motors (IMs) have also been considered for application as traction motors [10]. IMs have several merits such as low manufacturing costs and the high mechanical strength of their rotors, since there are no permanent magnets in their rotors. Moreover, high efficiency can be expected by using more flexible and better flux weakening control compared with PMSMs, since the field flux of the rotor can be changed with the stator current [11]. However, the efficiency under high load conditions is much lower due to the generation of joule loss in the rotor. Therefore, an IM with a copper rotor cage has been studied with the aim of decreasing the joule loss in the rotor [12].

As mentioned above, PMSMs, SRMs, and IMs have advantages and disadvantages, and their performances as motors for automobiles have been compared [13,14]. However, most of the previous studies have evaluated the performance at the operating speed of a typical automobile motor in less than 15,000 $\text{min}^{-1}$. It is predicted that the maximum speed of automobile motors will increase in the future due to reductions in size and weight, and it is necessary to evaluate the characteristics of each motor in the higher speed range from both electrical and mechanical points of view. In this research, motors that achieve output powers of 85 kW and maximum speeds of 52,000 $\text{min}^{-1}$ are proposed as EV traction motors, and the study reveals which motor type is most suitable for the realization of the output power and the maximum speed in terms of mechanical and electrical characteristics. The PMSM, SRM, and IM are designed to achieve the performances required for use in a high-speed drive. The performances of these designed motors are evaluated using FEA, and the advantages and disadvantages of their use in a high-speed drive are clarified.

## 2. Target Performances and Design Flow

The specifications for EV traction motors are shown in Table 1, and the required speed–torque characteristics are shown in Figure 1. As shown in Table 1 and Figure 1, the output power is 85 kW, the maximum torque is 70 Nm, the base speed is 11,500 $\text{min}^{-1}$, and maximum speed is 52,000 $\text{min}^{-1}$. The maximum phase current, maximum DC voltage, and the current density are constant at 356 Arms, 365 V, and less than 15 A/mm$^2$, respectively. The stack length is a constant 100 mm, and the motor diameter can be changed by the design to be less than 200 mm. As shown in the specification, the maximum speed of 52,000 $\text{min}^{-1}$ is very high compared with a general EV traction motor. It is possible to greatly downsize a motor's volume using high-speed rotation while obtaining high output power. To achieve these specifications and the smallest motor volume, the PMSM, SRM, and IM are designed, and the performances such as maximum torque, maximum output power, loss, and efficiency are evaluated at each rotation speed.

**Table 1.** Target performances.

| | |
|---|---|
| Output power [kW] | 85 |
| Maximum torque [N m] | 70 |
| Voltage source [V] | 365 |
| Maximum current [Arms] | 356 |
| Current density [A/mm$^2$] | 15 |
| Maximum diameter [mm] | 200 |
| Stack length of core [mm] | 100 |

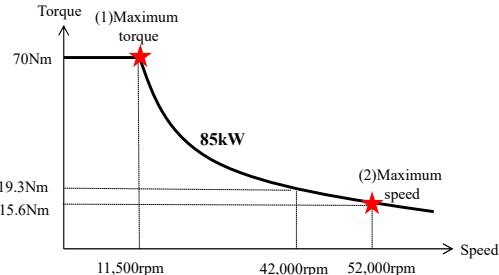

**Figure 1.** Target speed–torque characteristic.

Figure 2 shows the design flow. As shown in Figure 2, the design flow is separated into three parts, including the design of mechanical strength, the design of electric characteristics, and evaluation of motor performance. In the design of mechanical strength, the diameter and the shape of rotor to achieve enough mechanical strength with the maximum speed of 52,000 min$^{-1}$, which is determined using centrifugal force analysis. In the design of the electric characteristics, the number of turns, the shape of the stator, and the pole number are determined to obtain the maximum torque of 70 Nm and to retain the induced voltage as less than the DC voltage. In this study, PMSM and IM are assumed to be driven sinusoidally by the vector control with a sensor in all speed region, while SRM is assumed to be driven by hysteresis control at the low-speed region and voltage single pulse control at high-speed region. Actually, the driving by the sinusoidal wave in the high-speed region requires the high switching frequency in the inverter. However, the purpose of this paper is to evaluate the motor characteristics under ideal conditions, and the effects of drive conditions of the inverter are not considered.

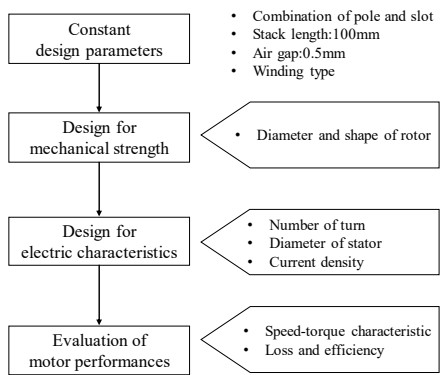

**Figure 2.** Design flow.

## 3. Design of PMSM

In this section, the specifications of the designed PMSM and the design process used to realize the required performances are described. Figure 3 shows the dimension of the designed PMSM, and Table 2 shows the specifications. As shown in Figure 3 and Table 2, the SPMSM is selected in this design. In general, the IPMSM has been widely used as traction motors since high output torque and a wide speed range can be obtained by

utilizing not only the magnet torque but also the reluctance torque. On the other hand, the SPMSM is more appropriate in terms of the mechanical strength compared with the IPMSM since the rotor shape of the SPMSM can reduce the Mises stress generated in the rotor more effectively than that of the IPMSM. The combination of the pole and slot is 4/6, and the diameter of the stator and the stack length are 174 mm and 100 mm. In addition, the permanent magnet is axially segmented by 16 layers to decrease the eddy current loss, and the permanent rotor magnet is assumed to be reinforced with a 0.2 mm retaining sleeve that is made of carbon-fiber-reinforced plastic (C-FRP), which has a high electrical resistivity of $1.5 \times 10^{-5}$ Ωm. The PMSM is assumed to be driven by a sinusoidal wave. The detailed design process is stated as follow.

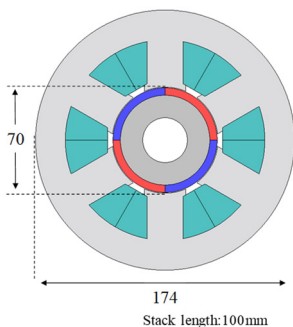

**Figure 3.** Dimensions of designed PMSM.

**Table 2.** Specifications of PMSM.

| | |
|---|---|
| Number of poles | 4 |
| Stator slots | 6 |
| Number of turns [turn/slot] | 7 |
| Resistance of winding [Ω] | 0.0018 |
| Winding type | Concentrated |
| Air gap [mm] | 0.5 |
| Magnet thickness [mm] | 5 |
| Current density [A/mm$^2$] | 10 |
| Material of magnetic steel sheet | 35H230 |
| Material of permanent magnet | NEOMAX-42 |
| Number of magnet segmentations | 16 |

### 3.1. Mechanical Design of PMSM

The Mises stress generated in the rotor at the maximum speed of 52,000 min$^{-1}$ is evaluated using FEA. JMAG-designer is used as the simulation tool. Figure 4 shows the comparison of Mises stress in the IPMSM and SPMSM. The two types of magnet arrangements (Model A and Model B) in the IPMSM and SPMSM, which are designed with constant rotor diameters of 70 mm, are considered. Surface permanent magnet type and interior permanent magnet type are abbreviated as SPM type and IPM type, respectively. For the magnet and core, sintered magnet (NEOMAX-42) and magnetic steel sheet (35H230) materials are used. Then, the Young's modulus and Poisson ratio of the magnetic steel sheet are 210,000 MPa and 0.3, respectively. Those of the magnet are 120,000 MPa and 0.3, respectively. As shown in Figure 4, in Model A and Model B of the IPM type, the high Mises stress is concentrated at the edge of flux barrier. On the other hand, the Mises stress of the SPM type is concentrated at the surface permanent magnet on the rotor, and it is much lower than that of the IPM type. Although these models of IPM types are not optimized to reduce the Mises stress, it is obvious that SPM types are more suitable for decreasing Mises stress compared with IPM types. Figure 5a shows Mises stress distribution for the rotor diameter in the SPM type. As shown in Figure 5a, the Mises stress can be reduced by decreasing the rotor diameter. Figure 5b shows the maximum Mises stress for the rotor

diameter. As shown in Figure 5b, considering the limit of the yield stress of 300 MPa, the rotor diameter is determined to be less than 70 mm. Moreover, the rotor magnet can be reinforced with a retaining sleeve made of carbon-fiber-reinforced plastic (C-FRP).

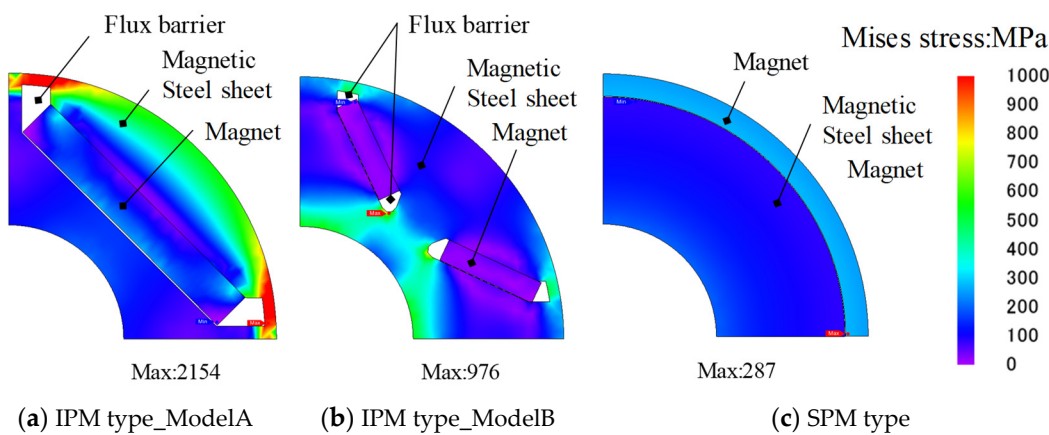

(**a**) IPM type_ModelA　　　(**b**) IPM type_ModelB　　　(**c**) SPM type

**Figure 4.** Mises stress in the rotor for IPM type (Model A and Model B) and SPM type.

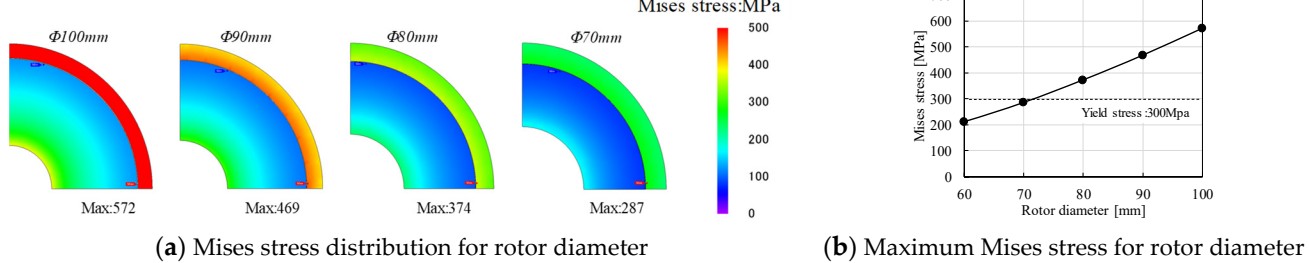

(**a**) Mises stress distribution for rotor diameter　　(**b**) Maximum Mises stress for rotor diameter

**Figure 5.** Mises stress of SPMSM at 52,000 min$^{-1}$.

### 3.2. Electrical Design of PMSM

In the electrical design, the number of turns is determined to achieve the maximum torque of 70 Nm and to maintain the induced voltage at less than half of the DC voltage in each rotation speed. The torque equation is expressed as follows:

$$T = p\psi_d i_q \tag{1}$$

where $T$, $p$, $\psi_d$, and $i_q$ are output torque, the number of pole pairs, magnet flux linkage, and the q-axis current, respectively. Additionally, the magnet flux linkage is expressed as follows:

$$\psi_d = N\phi_d \tag{2}$$

where $N$ is number of turns, and $\phi_d$ is magnet flux linkage in one turn/slot. From Equations (1) and (2), the relationship between the output torque and the number of turns is shown in Figure 6. Then, there are four pole pairs, and the phase current is constant at 356 Arms. As shown in Figure 6, the maximum torque of 70 Nm is obtained in more than six turns/slot.

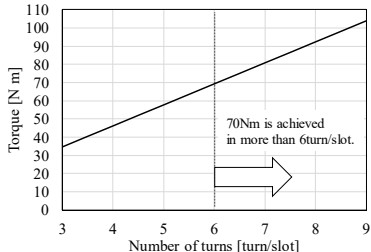

**Figure 6.** Relationship between output torque and number of turns.

In addition, the phase voltage has to be limited to less than half of the DC voltage. The condition of phase voltage is expressed as follows:

$$\sqrt{\frac{2}{3}}\sqrt{v_d{}^2 + v_q{}^2} < \frac{V_{dc}}{2} \tag{3}$$

where $v_d$, $v_q$, and $V_{dc}$ are the d-axis voltage, q-axis voltage, and DC voltage source, respectively. Then, the inductance condition to achieve the base speed and the output power is given by:

$$L_b \leq \frac{\sqrt{\frac{3}{8}\left(\frac{V_{dc}}{\omega_b}\right)^2 - \psi_d{}^2}}{i_q} \tag{4}$$

where $\omega_b$ and $L_b$ are the base angular velocity and inductance (there is a relationship $L_b = L_d = L_q$ due to SPMSM). In Equation (4), the maximum torque per ampere control is assumed under base speed; therefore, the d-axis current is 0 A. Moreover, the inductance condition to achieve the maximum speed and the output power is given by:

$$\omega_m\sqrt{\frac{2}{3}}\sqrt{(L_b i_q)^2 + (-L_b i_d + \psi_d)^2} < \frac{V_{dc}}{2} \tag{5}$$

Using Equation (5), the inductance condition is derived as follows:

$$L_b = \frac{-b \pm \sqrt{b^2 - 4ac}}{2a} < 0, \; L_1 < L_b < L_2. \tag{6}$$

Then, parameters *a*, *b*, *c* are as follows:

$$\begin{aligned} a &= I_m{}^2 \\ b &= -2\psi_d i_d \\ c &= \psi_d{}^2 - \frac{3}{8}\left(\frac{V_{dc}}{\omega_m}\right)^2 \end{aligned} \tag{7}$$

where $\omega_m$, $I_m$, and $i_d$ are the maximum angular velocity, the current vector amplitude, and the d-axis current, respectively. The functions of *a*, *b*, and *c* can be calculated using the parameters of a phase current of 356 Arms and a d-axis current at a beta angle of 78 degrees. From Equations (4) and (6), the inductance range of $L_b$ to achieve the condition of phase voltage at the base speed and maximum speed is shown in Figure 7. As shown in Figure 7, the number of turns, seven turns/slot, which satisfied Equations (4) and (6), is selected. A stator diameter that retained the slot area is used, which achieves less than 15 A/mm$^2$ of the current density.

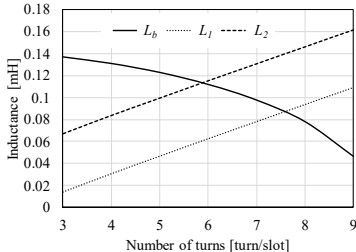

**Figure 7.** Inductance condition which achieves the demand torque at base speed 11,500 min$^{-1}$ and maximum speed 52,000 min$^{-1}$.

### 3.3. Torque and Phase Voltage at Required Speed of PMSM

The performances of PMSM are evaluated using FEA. Figures 8 and 9 show the torque waveforms and phase voltage waveform at the operation speed of 11,500 min$^{-1}$ and 52,000 min$^{-1}$, respectively. As shown in Figure 8a, the designed PMSM can achieve the maximum torque of 70 Nm at 11,500 min$^{-1}$. As shown in Figure 9b, the phase voltage is also suppressed to less than half of the DC voltage of 365 V at 52,000 min$^{-1}$, and the output power of 85 kW is obtained. However, a lot of the d-axis current is energized by the flux weakening control in 52,000 min$^{-1}$ to cancel out the magnet flux and to sustain the induced voltage.

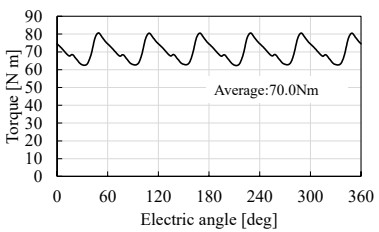 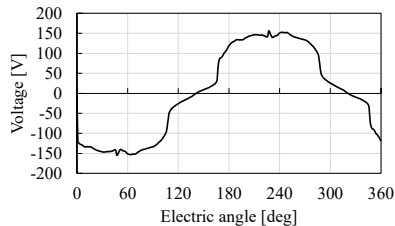

(a) Torque waveform (b) Voltage waveform

**Figure 8.** Torque waveform and phase voltage waveform in rotation speed of 11,500 min$^{-1}$, input current of 356 Arms, and beta angle of 0 degrees.

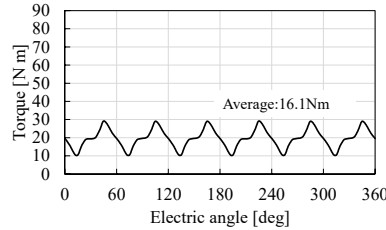 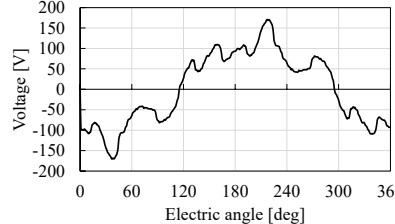

(a) Torque waveform (b) Voltage waveform

**Figure 9.** Torque waveform and phase voltage waveform in rotation speed of 52,000 min$^{-1}$, input current of 356 Arms, and beta angle of 78 degrees.

## 4. Design of SRM

In this section, the specifications of the designed SRM and the design process used to obtain the required performances are described. Figure 10 shows the dimensions of the designed SRM, and Table 3 shows the specifications of the SRM. As shown in Figure 10 and Table 3, the 8 poles and 12 slots are selected, and the motor diameter and stack length are 200 mm and 100 mm, respectively. The detailed design process is stated as follow.

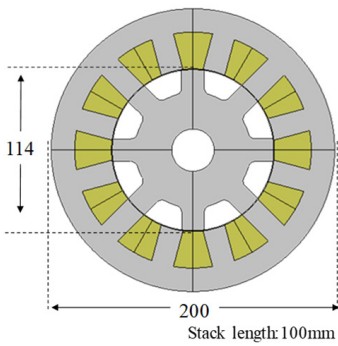

**Figure 10.** Dimension of designed SRM.

**Table 3.** Specifications of SRM.

| | |
|---|---|
| Number of poles | 8 |
| Stator slots | 12 |
| Number of turns [turn/slot] | 5 |
| Resistance of winding [$\Omega$] | 0.003 |
| Winding type | Concentrated |
| Air gap [mm] | 0.5 |
| Magnet thickness [mm] | 5 |
| Current density [A/mm$^2$] | 12.7 |
| Material of magnetic steel sheet | 35H230 |

*4.1. Mechanical Design of SRM*

Figure 11 shows the Mises stress distribution for the rotor diameter at the maximum speed of 52,000 min$^{-1}$. As shown in Figure 11a, high Mises stress is generated around the shaft and the edge of the teeth. As shown in Figure 11b, considering the limit of the Mises stress is under 300 MPa, the rotor diameter has to decrease to less than 114 mm.

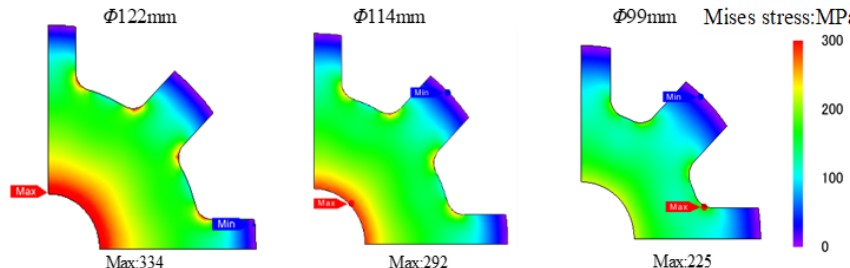

(a)  Mises stress distribution for the rotor diameter

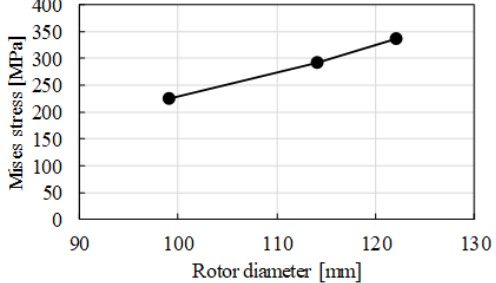

(b)  Maximum Mises stress for the rotor diameter

**Figure 11.** Mises stress of SRM in the rotor at 52,000 min$^{-1}$.

*4.2. Electrical Design of SRM*

In the electrical design, the number of turns is determined to achieve the maximum torque of 70 Nm. The torque of the SRM is generally expressed as follows:

$$T = \frac{P}{2}\frac{\partial L}{\partial \theta}i^2 \tag{8}$$

where $P$, $\partial L/\partial \theta$, and $i$ are number of poles in the rotor, the self-inductance variation, and phase current, respectively. In the SRM, the torque is proportional to the variation of self-inductance and the square of phase current. Equation (8) is rewritten as follows:

$$T = \frac{P}{2}\frac{\partial l}{\partial \theta}(Ni)^2 \tag{9}$$

where $\partial l/\partial \theta$ is the inductance variation in one turn/slot. As shown in Equation (9), the output torque is proportional to the inductance variation in one turn/slot and the square of the number of turns and the current. The inductance variation depends on the salient pole ratio of the rotor. As shown in Equation (9), the number of turns is adjusted to obtain the maximum torque of 70 Nm under the maximum current of 356 Arms. In general, SRMs are driven by a unipolar excitation current. Therefore, the number of turns is considered in the application of the ideal square waveform. Figure 12 shows the current waveform and torque waveform in five turns/slot. As shown in Figure 12a, the ideal phase current is 356 Arms, and the excitation is continued from the turn-on angle of 0 degrees to the turn-off angle of 90 degrees in one electrical period of 180 degrees. As shown in Figure 12b, it is confirmed that the output torque of 70 Nm can be obtained in five turns/slot. A stator diameter that retained the slot area is used, and a winding diameter under the coil factor of 0.6 is used, which achieves less than 15 A/mm$^2$ of the current density.

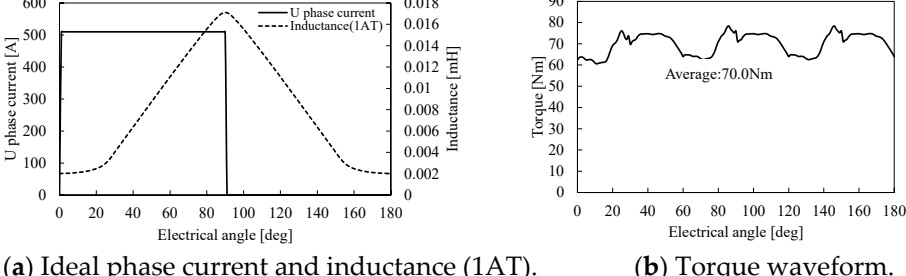

(**a**) Ideal phase current and inductance (1AT).      (**b**) Torque waveform.

**Figure 12.** Torque waveform for applying ideal phase current.

*4.3. Torque and Phase Voltage at Required Speed of SRM*

Using the drive method in the SRM, the current excitation started around the position where the inductance variation is positive by using the hysteresis control or the voltage single-pulse drive. Then, it has to be eliminated before the inductance variation becomes negative to avoid the generation of negative torque. From this drive method, the output current is changed by some parameters such as the DC voltage, inductance distribution, the turn-on angle, and the turn-off angle. In the performance evaluation, the turn-on angle and turn-off angle are determined under a constant input voltage of 365 V$_{dc}$ to achieve the required torque for each operation speed.

Figure 13 shows the current waveform, voltage waveform, and torque waveform at 11,500 min$^{-1}$. As shown in Figure 13, the hysteresis current control is used in 11,500 min$^{-1}$, and the output torque is about 70 Nm, while the phase current is 345 Arms. Figure 14 shows their waveforms at 52,000 min$^{-1}$. As shown in Figure 14, the voltage single pulse control is used in 52,000 min$^{-1}$. In the high-speed region, the current can be raised, and the torque can be obtained by setting the turn-on angle to an early timing as rotation speed

increases. The turn-off angle is also set to a timing earlier than the aligned position of rotor and stator to avoid negative torque.

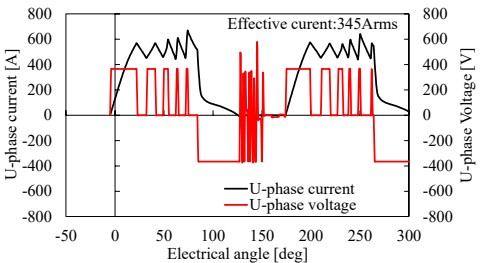

(**a**) Phase current and voltage waveforms.

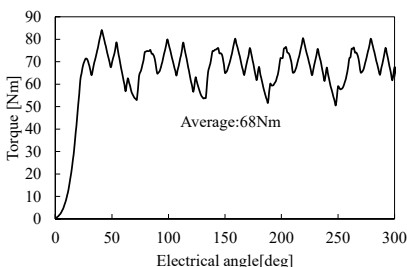

(**b**) Torque waveform.

**Figure 13.** Rotation speed of 11,500 min$^{-1}$, turn-on angle of −5 degrees, and turn-off angle of 85 degrees.

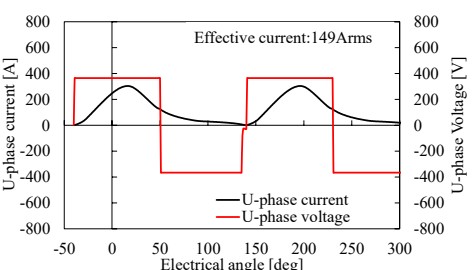

(**a**) Phase current and voltage waveforms.

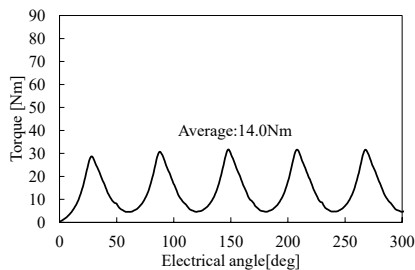

(**b**) Torque waveform.

**Figure 14.** Rotation speed of 52,000 min$^{-1}$, turn-on angle of −40 degrees, and turn-off angle of 51 degrees.

## 5. Design of IM

In this section, the specifications of the designed IM and the design process used to realize the required performances are described. Figure 15 shows the dimension of the designed IM, and Table 4 shows the specifications of the IM. As shown in Figure 15 and Table 4, 4 poles and 24 slots are used, and distributed winding is applied. The number of bars are 36, and the bars are made of aluminum. The motor diameter and stack length are 160 mm and 100 mm, respectively. The detailed design process is stated as follows.

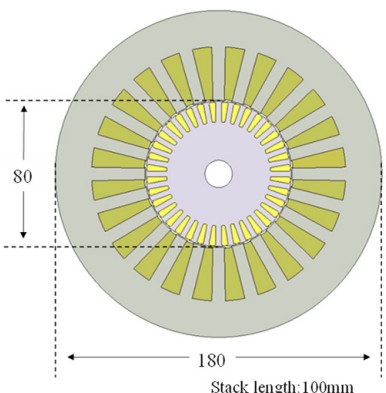

**Figure 15.** Dimension of designed IM.

**Table 4.** Specifications of IM.

| | |
|---|---|
| Number of poles | 4 |
| Stator slots | 24 |
| Number of bars | 36 |
| Number of turns [turn/slot] | 4 |
| Resistance of winding [Ω] | 0.003 |
| Winding type | Distributed |
| Air gap [mm] | 0.5 |
| Current density [A/mm$^2$] | 11 |
| Material of magnetic steel sheet | 35H230 |
| Material of bar | Aluminum |

*5.1. Mechanical Design of IM*

Figure 16 shows the Mises stress distribution for the rotor diameter at the maximum speed of 52,000 min$^{-1}$. As shown in Figure 16, high mises stress is generated around the shaft. Considering the limit of Mises stress is under 300 MPa, the rotor diameter has to decrease to less than 80 mm. Therefore, the rotor diameter is set to 80 mm.

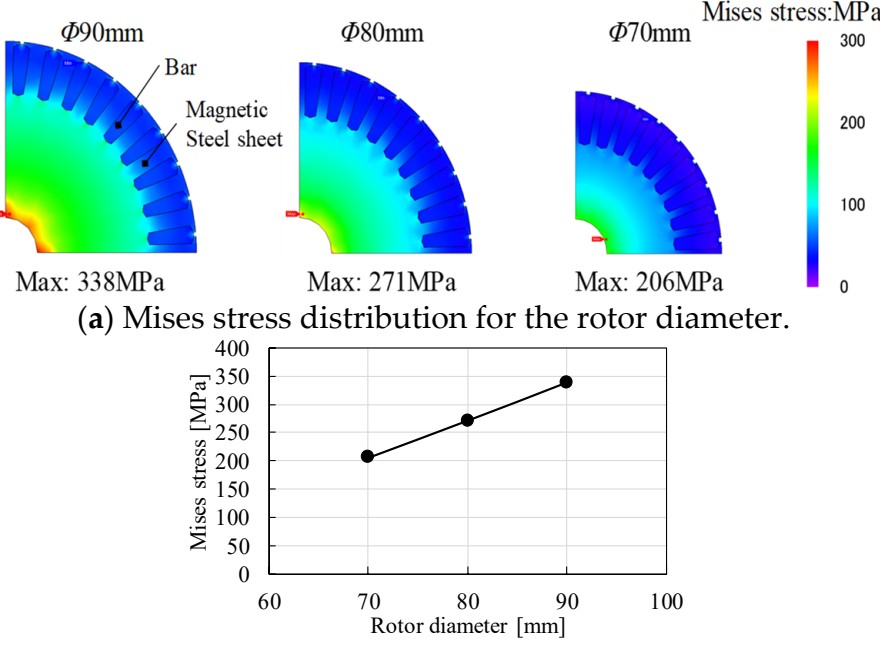

(**a**) Mises stress distribution for the rotor diameter.

(**b**) Maximum Mises stress for the rotor diameter.

**Figure 16.** Mises stress of IM in the rotor at 52,000 min$^{-1}$.

*5.2. Electrical Design of IM*

In the electrical design, the number of turns is determined using FEA to achieve the required maximum torque of 70 Nm. Figure 17 shows the frequency–torque characteristics for the number of turns. The electrical frequency is changed under the locked rotor condition. As shown in Figure 17, the number of turns changed from one turn/slot to four turns/slot, and the required maximum torque of 70 Nm is obtained using four turns/slot. Then, machine parameters such as the secondary resistance, the primary leakage inductance, the secondary leakage inductance, and the mutual inductance are obtained by carrying out the locked test and the no-load test using FEA. The machine parameters are shown in Table 5.

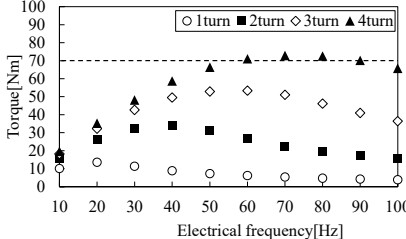

**Figure 17.** Torque characteristics for number of turns of designed IM.

**Table 5.** Machine parameters of IM.

| | |
|---|---|
| Secondary resistance [$\Omega$] | 0.071 |
| Primary leak inductance [$\mu$H] | 26.2 |
| Secondary leak inductance [$\mu$H] | 26.2 |
| Mutual inductance [$\mu$H] | 278 |

*5.3. Torque and Phase Voltage at Required Speed of IM*

The performances of the IM are evaluated using FEA. Figures 18 and 19 show the torque waveforms and phase voltage waveform at the operation speed of 11,500 min$^{-1}$ and 52,000 min$^{-1}$, respectively. As shown in Figure 18, the designed IM can achieve the maximum torque of 70 Nm under conditions with a phase current of 356 Arms and slip of 0.16. However, as shown in Figure 17, the output torque is 8.1 Nm, the required torque of 15.6 Nm and 85 kW cannot be obtained at 52,000 min$^{-1}$. The phase voltage is more than 200 V, and the voltage source required is 420 $V_{dc}$.

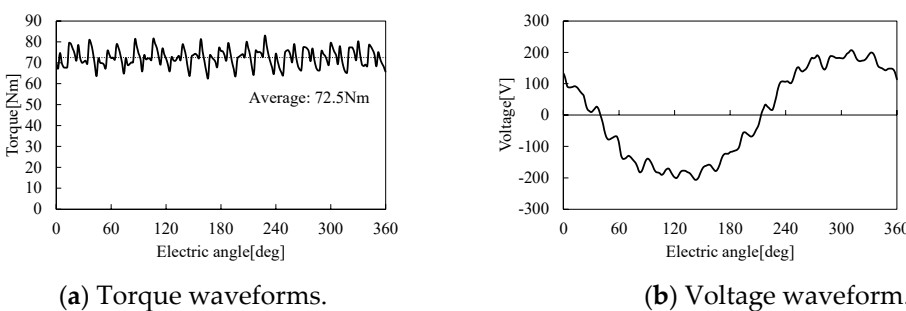

(**a**) Torque waveforms.  (**b**) Voltage waveform.

**Figure 18.** Rotation speed of 11,500 min$^{-1}$, input current of 356 Arms, and slip of 0.16.

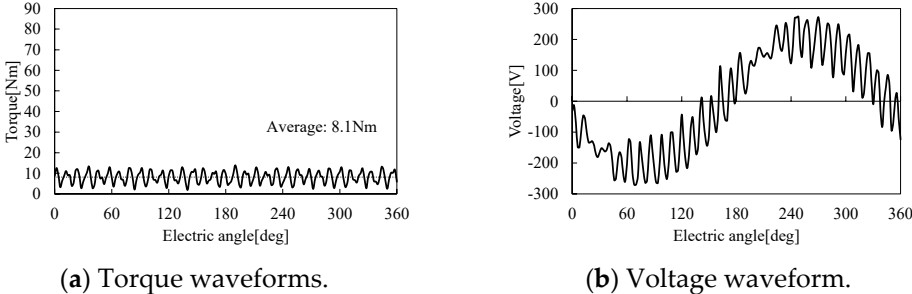

(**a**) Torque waveforms.  (**b**) Voltage waveform.

**Figure 19.** Rotation speed of 52,000 min$^{-1}$, input current of 159 Arms, and slip of 0.096.

**6. Performance Comparison of Designed Motors**

In this section, the motor performances of the motor volume, the mechanical strength, the output power, the loss, and efficiency of the designed PMSM, SRM, and IM are compared, and the results clarify which motor is most suitable for use as a high-speed traction motor.

### 6.1. Motor Volume

Table 6 shows the comparison of motor volume. As shown in Table 6, the comparison is based on the SRM volume of 1.0 p.u. The PMSM and IM are smaller than the SRM in terms of volume. The rotor diameter is determined depending on the limitation of mechanical strength at the maximum speed, and the stator diameter is determined by the slot area which can achieve the condition of the current density and ensure the number of turn and the current to obtain the maximum torque. The PMSM has the smallest motor volume, since the slot area can be easily secured by the combination of poles/slots and concentrated winding and the required torque can easily be obtained using the magnet torque.

**Table 6.** Comparison of motor volume.

|  | PMSM | SRM | IM |
|---|---|---|---|
| Size [mm] | $\varphi 174 \times L100$ | $\varphi 200 \times L100$ | $\varphi 180 \times L100$ |
| Rotor outer diameter [mm] | 70 | 114 | 80 |
| Motor volume [p.u.] | 0.76 | 1.0 | 0.81 |
| Current density [A/mm$^2$] | 10 | 12.7 | 11 |

### 6.2. Mechanical Strength

Figure 20 shows the comparison of maximum Mises stress for the rotor diameter. As shown in Figure 20, the rotor diameters of PMSM, SRM, and IM that achieve yield stress less than 300 MPa are 70 mm, 114 mm, and 80 mm, respectively. The Mises stress in the SRM is maintained at a low value for increasing of the rotor diameter since the rotor structure consists of a magnetic steel sheet alone. That is, it is confirmed that the SRM has the highest mechanical strength.

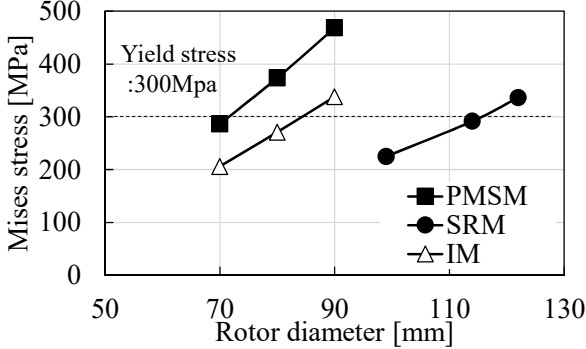

**Figure 20.** Comparison of Mises stress at the maximum speed of 52,000 min$^{-1}$.

### 6.3. Output Characteristics

Figure 21 shows the output power characteristics. As shown in Figure 21, the PMSM realizes the maximum torque of 70 Nm and the output power of 85 kW at 11,500 min$^{-1}$ and 52,000 min$^{-1}$, although the magnet flux has to be strongly weakened by flux weakening control to suppress the induced voltage at the maximum rotational speed of 52,000 min$^{-1}$. In the SRM, the output torque is 67.8 Nm at 11,500 min$^{-1}$, and it is 14.2 Nm at 52,000 min$^{-1}$. Although the output power is slightly lower than the required torque for the output power of 85 kW, the required torque can be obtained by optimizing the turn-on angle and turn-off angle. In the IM, the output torque is 71 Nm at 11,500 min$^{-1}$, and it achieves the required maximum torque. However, the output torque at 52,000 min$^{-1}$ is much lower than the required torque for the output power of 85 kW, since the voltage is limited by increasing the rotor flux due to the second current.

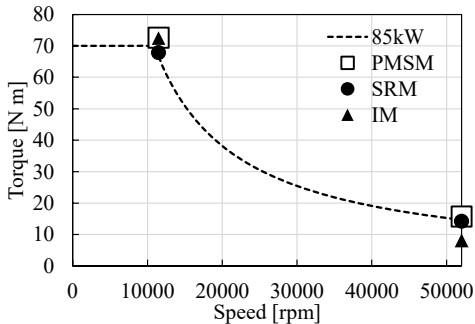

**Figure 21.** Output power characteristics.

### 6.4. Loss and Efficiency

The loss and efficiency at the base speed of 11,500 min$^{-1}$ and the maximum speed of 52,000 min$^{-1}$ are evaluated. Figure 22 shows losses of joules in the rotor and stator and core loss. As shown in Figure 22, the joule loss and the core loss in the PMSM are low, since the input current and the number of turns are small under the small-sized motor volume, owing to utilization of the permanent magnet. On the other hand, the joule loss in the stator of the PMSM is higher than that of the SRM at 52,000 min$^{-1}$, since a lot of d-axis current is needed by the flux weakening control. The SRM also achieves low levels of joule loss in the stator, since the input current and number of turns are almost same for the PMSM instead of the rotor diameter increasing. On the other hand, the core loss increases compared with the PMSM. The IM generates high levels of joule loss, since the joule loss in the rotor accounts for a large percentage (the stator joule loss and the rotor joule loss are 1.1 kW and 23 kW at 11,500 min$^{-1}$, respectively, and the stator joule loss and the rotor joule loss are 0.23 kW and 7.8 kW at 52,000 min$^{-1}$, respectively). The core loss of the IM is also higher than that of the PMSM.

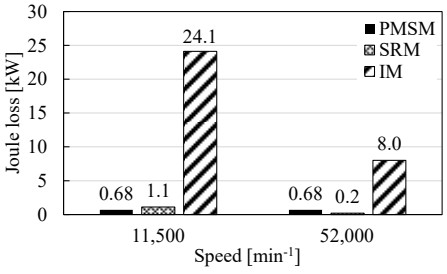

(**a**) Joule loss in rotor and stator.

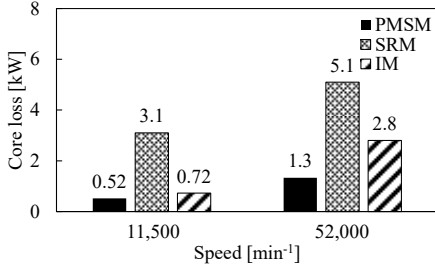

(**b**) Core loss.

**Figure 22.** Comparison of joule loss and core loss.

Figure 23 shows the eddy current loss of the magnet and retaining sleeve for the PMSM. As shown in Figure 23a, it can be seen that the increase in eddy current loss of the magnet in the high-speed rotation is serious in the PMSM. The number of magnet layers needed to be increased to reduce the eddy current loss. In the design, the magnet eddy current loss is suppressed by dividing the magnet into 16 layers. However, a further reduction in eddy current loss is required to avoid thermal demagnetization in the magnet. As shown in Figure 23b, the eddy current loss also increased in the retaining sleeve of the magnet in high-speed rotation. Therefore, it is necessary to use a material with high electrical resistance and thinness for the retaining sleeve. In the design, eddy current loss is suppressed by using the C-FRP as the reinforcement material, which has a high electrical resistivity of $1.5 \times 10^{-5}$ Ω m. Actually, the number of layers and the material of retaining sleeve should be determined in consideration of the trade-off with manufacturing cost.

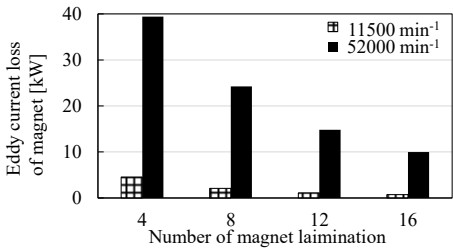    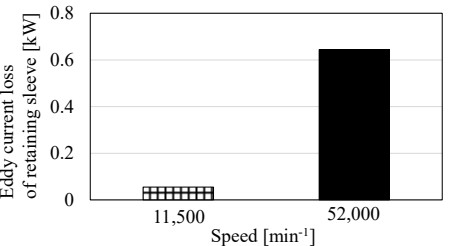

(**a**) Eddy current loss of magnet.       (**b**) Eddy current loss of retaining sleeve.

**Figure 23.** Eddy current loss of PMSM.

Figure 24 shows the windage loss. The windage loss is calculated by the following equation [15]:

$$W_a = K\pi C_d \rho R^4 \omega_h^3 L \tag{10}$$

where $W_a$, $K$, $C_d$, $\rho$, $R$, and $L$ are the windage loss, salient-pole correction factor, skin friction coefficient, air density, radius of high-speed rotor, and motor stack length, respectively. In the SRM, the windage loss increases considerably at 52,000 min$^{-1}$ due to the salient-pole structure of the rotor. The windage loss can be reduced by using the shroud or cylindrical rotor structure [16]. Figure 25 shows the comparison of the efficiency. As shown in Figure 25, the PMSM can achieve the highest efficiency of 97% at 11,500 min$^{-1}$ and 86% at 52,000 min$^{-1}$. The SRM achieves the efficiency of 95% at 11,500 min$^{-1}$ and 81% at 52,000 min$^{-1}$. The efficiencies of the IM are 77.8% at 11,500 min$^{-1}$ and 80.3% at 52,000 min$^{-1}$.

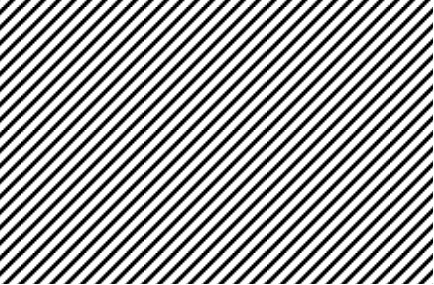

**Figure 24.** Windage loss.

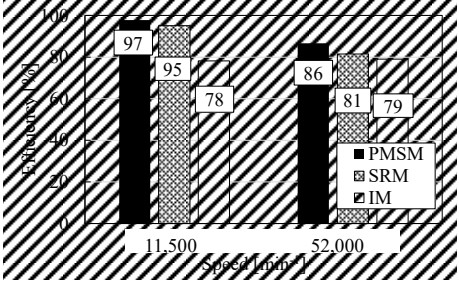

**Figure 25.** Motor efficiency.

## 7. Conclusions

In this paper, the type of motor suitable for use as a high-speed motor in EV traction applications was clarified in terms of the motor volume, mechanical strength, output power, loss, and efficiency. The PMSM, SRM, and IM were designed with the aim of achieving an output power of 85 kW, a maximum torque of 70 Nm, and a maximum speed of 52,000 min$^{-1}$, and the performances were evaluated using FEA. Table 7 shows the comparison of these motor performances. As shown in Table 7, the PMSM was advantageous in terms of the downsizing of the motor volume and motor efficiency. However, it is necessary to reduce the eddy current loss of the magnet at 52,000 min$^{-1}$ by increasing the number

of magnet layers and reducing the harmonic flux by applying distributed winding. They have to be designed considering the trade-off between increased cost and larger motor size. On the other hand, the SRM was advantageous in terms of the high mechanical strength of the rotor, and it is suitable for high-speed rotation. The motor volume of the SRM was larger than that of the PMSM. Although the efficiency of the SRM was lower than that of the PMSM at the high load condition of 11,500 $\mathrm{min}^{-1}$, the efficiency at 52,000 $\mathrm{min}^{-1}$ can be improved by decreasing windage loss using the shroud or cylindrical rotor structure. In this study, the efficiency of the IM at 11,500 $\mathrm{min}^{-1}$ was considerably lower due to the joule loss in the rotor. On the other hand, the IM had the advantage of no magnet eddy current loss, and lower iron loss and wind loss than the SRM, so it has potential as a high-speed traction motor if the joule loss can be reduced.

**Table 7.** Motor performances of high-speed drive.

|  | Motor Size | Mises Stress at Maximum Speed | Motor Efficiency |
| --- | --- | --- | --- |
| PMSM | $\Phi$174 mm $\times$ $L$100 mm | 287 MPa (Rotor diameter: 70mm) | 11,500 $\mathrm{min}^{-1}$: 97% 52,000 $\mathrm{min}^{-1}$: 86% |
| SRM | $\Phi$200 mm $\times$ $L$100 mm | 291 MPa (Rotor diameter: 114mm) | 11,500 $\mathrm{min}^{-1}$: 95% 52,000 $\mathrm{min}^{-1}$: 81% |
| IM | $\Phi$180 mm $\times$ $L$100 mm | 258 MPa (Rotor diameter: 80mm) | 11,500 $\mathrm{min}^{-1}$: 78% 52,000 $\mathrm{min}^{-1}$: 79% |

**Author Contributions:** Conceptualization, K.A. (Kan Akatsu); Investigation, K.A. (Kohei Aiso); Writing—original draft, K.A. (Kohei Aiso). All authors have read and agreed to the published version of the manuscript.

**Funding:** This research received no external funding.

**Institutional Review Board Statement:** Not applicable.

**Informed Consent Statement:** Not applicable.

**Data Availability Statement:** Not applicable.

**Conflicts of Interest:** The authors declare no conflict of interest.

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
