# Peer review of "Performance Comparison of High-Speed Motors for Electric Vehicle"

_wevj, doi:10.3390/wevj13040057_

Round 1

Reviewer 1 Report

The paper is well written, smooth and interesting. It offers many points of reflection and can be useful for advances in the field.
Perhaps, if desired, the authors could give indications regarding the further development of the research (other powers, other speeds, limits beyond which one cannot go, etc.).

Author Response

Thank you for your depth review. The paper was revised by using the language editing service of MDPI.

Reviewer 2 Report

The paper are proposing a work "Comparison of Performance of High Speed Motors for Electric Vehicles" that studies three types of motors that have a power of 85kW. This paper is designed for high-speed motor type is suitable. The paper is well illustrative and with excellent results, very well presented and the subject is of interest to the WEVJ. Therefore, I recommend the paper in its current format.

Author Response

(The authors gave the same response as above.)

Reviewer 3 Report

  1. In the paper the comparison of three types of electrical motors: PMSM, SRM and IM for application in high-speed drive system of electric vehicle is performed. The considerations give the impression of a certain limitation of the depth of consideration of the issue. The analysis was mainly based on considering the values of Mises stresses for particular designs of motors. It seems advisable to deepen the analysis by introducing studies with using mathematical optimization techniques.
  2. The article lacks a more detailed description of the structure of the drive system of electric vehicle. The topology of the converter system for motor control, the range of the output frequency, the motor control algorithm, the sensors and feedback used in control system and the structure of the mechanical system were  not described.
  3. The text of the article requires grammatical corrections. There are incorrect grammar expressions in many places. There are often inconsistencies between the subject and the verb in sentences.
  4. The abstract of the article is too short and does not include all the issues discussed in the article.
  5. There are some of editorial anomalies in the article that should be corrected for obtaining the better form of the article. They were presented in detail below.
  6. The Abstract, L.9-12. This sentence is too long, it includes more than 55 words. The use of long sentences is not recommended in abstracts which are usually quickly read.
  7. For the PMSM the number of pole pairs is given in Table 2, but for SRM and IM the number of poles is given as data in Tables 3 and 4. In Eq.(1) small letter p is used, but in Eq.(8) and (9) the capital letter P is used.
  8. The abbreviations: SPM and IPM on page 3 should be explained.
  9. The References should bi cited for the Eq.(8).
  10. The parameters of IM given in Table 5 should be verified. It is not stated, whether rotor parameters are transferred to the stator side. Doubts are raised by the excessively small value of the mutual inductance compared to the leakage inductances.

Author Response

Thank you for your depth review. Our answers to the questions and comments are attached.
